# Predicting subsurface sonar observations with satellite-derived ocean surface data in the California Current Ecosystem

**Kellie R. Gadeken**[1], **Maxwell B. Joseph**[2]*, **Joseph McGlinchy**[2], **Kristopher B. Karnauskas**[3,4], **Carrie C. Wall**[4,5]

**1** Bredesen Center for Interdisciplinary Research and Graduate Education, University of Tennessee, Knoxville, Tennessee, United States of America, **2** Earth Lab, University of Colorado Boulder, Boulder, Colorado, United States of America, **3** Atmospheric and Oceanic Sciences, University of Colorado Boulder, Boulder, Colorado, United States of America, **4** Cooperative Institute for Research in Environmental Sciences, University of Colorado Boulder, Boulder, Colorado, United States of America, **5** National Centers for Environmental Information, National Oceanographic and Atmospheric Administration, Boulder, Colorado, United States of America

* maxwell.b.joseph@colorado.edu

**Data Availability Statement:** The raw data are available on Figshare: https://figshare.com/articles/Sonar_data_from_Gadeken_et_al_2020/12397958.

## Abstract

Vessel-based sonar systems that focus on the water column provide valuable information on the distribution of underwater marine organisms, but such data are expensive to collect and limited in their spatiotemporal coverage. Satellite data, however, are widely available across large regions and provide information on surface ocean conditions. If satellite data can be linked to subsurface sonar measurements, it may be possible to predict marine life over broader spatial regions with higher frequency using satellite observations. Here, we use random forest models to evaluate the potential for predicting a sonar-derived proxy for subsurface biomass as a function of satellite imagery in the California Current Ecosystem. We find that satellite data may be useful for prediction under some circumstances, but across a range of sonar frequencies and depths, overall model performance was low. Performance in spatial interpolation tasks exceeded performance in spatial and temporal extrapolation, suggesting that this approach is not yet reliable for forecasting or spatial extrapolation. We conclude with some potential limitations and extensions of this work.

## Introduction

Sonars that focus on the water column, the volume of ocean from the near surface to near the seabed, are used widely in fisheries science and management. The backscatter (or acoustic reflectance) from these sonars help characterize the distribution of marine life beneath the surface, from zooplankton to large predatory fish, by estimating biomass [1], trophic- and species-level identification [2, 3], and measuring school size and behavior [4–6]. This information can be used to understand ecosystem dynamics and inform stock assessments. However, vessel-based sonar data are expensive to collect, and have limited spatio-temporal coverage due to the constraints of the time and location of research cruises. The depth to which a water-

**Funding:** CB and KK received funding for this work through the University of Colorado Boulder Cooperative Institute for Research in Environmental Science (CIRES) Innovative Research Program (IRP): https://cires.colorado.edu/about/institutional-programs/innovative-research-program. This work was also supported by the University of Colorado Boulder Grand Challenge's investment in Earth Lab. The funders had no role in the study design, data collection and analysis, decision to publish, or preparation of the manuscript.

**Competing interests:** The authors have declared that no competing interests exist.

column sonar can survey depends largely on the transducer frequency and pulse length. In general, lower frequency transducers (like 18 kHz) can record to depths exceeding 1000 m while higher frequency transducers (like 200 kHz) can only record to about 150 m depth [7]. Most shipboard transducers are also unable to measure the upper 5-10 m of the water column because of draft (the depth of a ship's hull below the waterline), transducer ringdown, and other factors. Conversely, satellite-derived oceanographic data products provide information on surface (or to 1 optical depth) ocean conditions multiple times per day at a horizontal resolution down to 250 m. These data potentially provide insight into surface manifestations of complex physical and biological processes.

If satellite data can predict subsurface biomass measured by sonar, it may be possible to predict marine life over broader spatial regions with higher frequency using satellite observations. The first step to such predictive capabilities is to evaluate relationships between satellite observations and acoustic returns. This approach has been demonstrated for Pacific sardine (*Sardinops sagax*) in the California Coastal Ecosystem (CCE) [8]. The key challenge is to identify methodology to analyze these temporally and spatially diverse datasets for general biomass (non-species specific) distributions.

Several studies have evaluated the fidelity of satellite estimates of biologically relevant properties of the surface and water column. The 1% light level zone depth ($z_{1\%}$) is a measure of the depth where only 1% of the surface photosynthetic available radiation remains in the water column. The work done by [9] shows that in Case-1 waters (which have inherent optical properties adequately described by phytoplankton, and in turn, chlorophyll concentration) remotely sensed measurements of $z_{1\%}$ from optical satellite imaging systems show good agreement with *in situ* measurements across the visible wavelength region for depth ranges of approximately 4-80 m. More conservatively, $z_{10\%}$ showed similarly good agreement in the 20-30 m depth range. The authors in [9] also show that chloropyll-a concentrations estimated by remote sensing reflectances show good agreement with *in situ* measurements, particularly for waters with $z_{1\%}$ deeper than 30 meters.

The present study focuses on the CCE as a test region, motivated by several reasons. The CCE is a coastal upwelling biome in the eastern North Pacific Ocean, characterized by a highly productive and dynamic environment supporting planktonic crustaceans (euphausiids), a key food for large marine predators and a variety of commercially and environmentally critical fisheries. Aggregations of euphausiid species in the CCE, namely North Pacific krill (*Euphausia pacifica*), are typically found less than 30 m depth during the night and between 50-100 m depth during the day on the shelf and slope, and deeper offshore [10]. *Thysanoessa spinifera* is also very abundant in the CCE, with other species important at a local level. This diel vertical migration (daily movement to disperse in shallower waters at night and aggregate at depth during the day) is a well-known, predictable behavior driven by feeding and predator avoidance behavior. These patterns can also be easily discerned from water-column sonar data to effectively identify and map their distribution [7, 10–12]. Moreover, the CCE has been the focus of a large number of *in situ* [13], remote sensing [14], and modeling studies of coastal pelagic fish [15]. Coastal pelagic fish species in the CCE include jack mackerel (*Trachurus symmetricus*), Pacific mackeral (*Scomber japonicus*), and Pacific sardine (*Sardinops sagax*), which are most commonly found between 0-50 m depth but can extend down to 100 m (Pacific sardine) and 300 m (jack and Pacific mackeral) [16]. Northern anchovy (*Engraulis mordax*) and Pacific herring (*Clupea pallasii*) are distributed from 0-200 m depth [16]. Pacific hake (*Merluccius productus*) are found down to 600 m depth but most commonly between 200-500 m [16, 17]. Pacific saury (*Colalabis saira*) is also found in the CCE down to 250 m, though away from the coast and limited to Central and Northern California [16]. With the exception of Pacific hake, these species undergo diel vertical migration [16].

Here, we adopt a data-driven approach to explore whether satellite measurements of ocean surface characteristics can predict observations from water-column sonar data collected in the CCE over a 5 year period, which includes the 2015/16 El Nino. We use random forests to learn a functional mapping from satellite and physical explanatory variables to sonar data previously collected as part of National Oceanic and Atmospheric Administration (NOAA) National Marine Fisheries Service (Fisheries) and Fisheries and Oceans Canada (DFO) efforts to quantify Pacific Hake (*Merluccius productus*) biomass. We evaluate the ability of these models to interpolate and extrapolate in space and time using a combination of spatial and temporal out-of-sample predictive checks.

## Materials and methods

### Sonar data

Water-column sonar data were collected on NOAA Ship *Bell M. Shimada* by NOAA Northwest Fisheries Science Center (NWFSC) scientists to inform management decisions for Pacific Hake fisheries. NWFSC-DFO surveys ("Joint U.S.-Canada Integrated Acoustic Survey of Pacific Hake") are focused in the boreal summer (largely July–August) along the Pacific coast of the United States and Canada. These surveys use systematic transects that result in approximately overlapping cruise tracks between surveys at approximately the same time of year. A Simrad EK60 split-beam echosounder with 18 kHz, 38 kHz, 70 kHz, 120 kHz, and 200 kHz transducers was employed on NWFSC cruises conducted in 2011, 2012, 2013 and 2015 [18–21]. These data were archived at the NOAA National Centers for Environmental Information where they are publicly accessible and subsequently used in this analysis. The NWFSC 2014 cruise was not included in the analysis due to its sparse spatial coverage compared to the other four cruises.

Prior to each cruise, the echosounder was calibrated using the standard sphere method [7, 22]. Target strength and echo integration data were collected to calculate echosounder gain parameters to ensure the quality of the system performance. To minimize the effect of surface bubbles and transducer ringdown, acoustic data were collected from 10 m below the surface, roughly 5 m below the centreboard-mounted transducer face. Data were collected to a maximum depth of 750 m. With the transducer depth accounted for, the volume backscatter strength ($S_V$) sample depth is relative to the sea surface. This recording range resulted in a ping rate of 1 ping per 1.1 s.

All files were processed using Echoview (Myriax, 10) by first aligning pings in the time/distance domain across the frequency components. Data were then binned vertically to 1000 data points between 0 and 750 m (i.e., $S_V$ at 0.75 m intervals). Noise filters were applied to remove background noise and intermittent impulsive noise. Background noise was removed following [23] where the signal-to-noise ratio was set to 10 dB. Impulsive noise "spikes" were removed following [24] where a ping was removed if the $S_V$ was 10 dB higher or lower than the adjacent pings. Pings associated with transient noise and attenuated signal were removed following [24]. The transient noise algorithm identifies and adjusts sample values that are significantly higher than those of surrounding samples, namely 10 dB above a 5x9 sample window. The attenuated signal algorithm identifies pings that show decreased signal strength (10 dB) when compared to the 3 surrounding pings. Additional noise or data of questionable quality due to transmission loss were also removed. These parameters were based on empirical evidence by transducer frequency: 70 kHz data collected in depths beyond 500 m were removed, 120 kHz data were removed beyond 275 m depth, and 200 kHz data were removed beyond 150 m depth. Acoustic reflections from the seafloor were delineated using Echoview's "best bottom candidate" automated algorithm focused on a peak threshold of -30 dB to distinguish between

the water column and seafloor. Data were removed up to 5 m above the detected bottom to help account for imperfections in the seafloor detection.

The acoustic dataset resulted in over 33,000 files. As such, it was impractical to manually scrutinize the seafloor detection and completely remove all unwanted/non-biological acoustic signal. The above automated processes aimed to remove as much noise as possible in an efficient manner. However, the persistence of a fraction of the original noise is expected. By examining the data over a large scale, we anticipate the influence of any remaining noise to be insignificant compared to the overall signal.

After removing data associated with noise and the seafloor, the water-column sonar backscatter ($S_V$) was integrated into 4 km horizontal bins and vertical bins of 10-50 m depth ("shallow water"), 50-200 m ("mid water"), 200-500 m ("shelf water"), and 500-750 m ("deep water") for each frequency at a threshold of -80 dB. This undifferentiated integrated backscatter (acoustic energy represented by nautical area scattering coefficient ($s_A$, m$^2$ nautical mile$^{-2}$) [25], hereafter referred to as NASC, provides a proxy for biomass across the cruise track at depths that align with anticipated biological features (shallow water: epipelagic layer, near surface euphausiids and other zooplankton layers, small fish, and night-time distribution; mid water: end of epipelagic layer, overlap with euphausiids, and small fish, and possible hake distribution; shelf water: start of mesopelagic layer, some euphausiid sp. and typical depth for hake; and deep water: deep fish assemblages) [10, 11, 17, 26]. Due to the lack of classification of the backscatter, the results will not inherently distinguish between species or trophic levels. However, the examination of the variability of acoustic energy across the frequencies has been used to discern species [27, 28]. Size and certain characteristics (swim bladder vs. non-swimbladder) of marine organisms influence its frequency response curve (how strongly sound reflects at discrete frequencies). In general, large swim-bladdered fish reflect strongest at low frequencies (18 and 38 kHz) while small euphausiids reflect strongest at high frequencies (120 and 200 kHz) [3]. Myctophid fish (Myctophidae spp), present throughout the CCE do not follow these generalizations and, while small, likely reflect strongly at low frequencies [29]. The horizontal extent of the NASC values align with the satellite data described below.

## Satellite data

We acquired 4 km resolution chlorophyll-a concentration, normalized fluorescence line height, particulate organic carbon, and sea surface temperature (SST) from MODIS (Terra and Aqua) level-3 ocean color standard mapped image products. The satellite data followed the same spatial extent where the sonar data were collected and averaged using a 7 day window (+/- 3 days from the date of sonar data collection) using Google Earth Engine [30]. Observations from Terra and Aqua collected on the same day were averaged. We then averaged observations from the 7 day window to derive a mean value for each satellite variable at each 4 km sonar aggregation bin. An averaging window of 7 days minimizes the impact of missing MODIS observations, with the additional benefit of minimizing the impact of short, natural lags (less than 3 days) that may be expected between satellite measurements and sonar returns. Finally, we filtered the data to exclude sonar observations for which no satellite data were available. The optical column depth contributing to MODIS observations will vary with water column conditions, however, based on the work of [9] we expect the shallower depths of the sonar data to be most relevant to the satellite derived observations.

## Other explanatory variables

In addition to the oceanographic variables derived from MODIS satellite data, we used geographic and observation-level variables to help explain the sonar backscatter data (Table 1).

**Table 1. List of explanatory variables used in the random forest models.** For each variable, we list its name, type, source, and whether it was included in the reduced model. The full model includes all explanatory variables.

| Explanatory variable | Type | Source | In reduced model |
|---|---|---|---|
| Daytime | Binary | Sonar data | yes |
| Distance from shore | Real | Sonar data | yes |
| Frequency | Categorical | Sonar data | yes |
| NASC depth bin | Categorical | Sonar data | yes |
| Latitude | Real | Sonar data | yes |
| Ocean depth | Real | GEBCO | yes |
| Chlorophyll a | Real | MODIS | no |
| Normalized fluorescence line height | Real | MODIS | no |
| Particulate organic carbon | Real | MODIS | no |
| Sea surface temperature | Real | MODIS | no |

Geographic variables included distance from shore, and ocean depth derived from the general bathymetric chart of the ocean (GEBCO) 2014 30 arc-second grid bathymetry dataset [31]. Observation variables include characteristics associated with the sonar data collection, namely a binary indicator for whether the data were collected during the day or night, and categorical indicators for wavelength, depth bin, and wavelength-depth bin combinations.

## Random forests for sonar prediction

We used random forest models to predict NASC as a function of the explanatory variables. We considered two types of models. The "full" model included all explanatory variables. The "reduced" model included only the geographic and observation covariates, and no satellite covariates. We used 100 trees each, a minimal node size of five, and unlimited tree depth. Because the distribution of NASC values is heavy-tailed, and the random forest regression models used mean-squared-error loss, all models were trained to predict NASC on a $\log_e$ scale. By comparing the performance of these models, we can evaluate whether satellite data are useful for predicting acoustic biomass.

## Model evaluation

We used two strategies to evaluate out-of-sample predictive performance. First, to evaluate predictive performance in unsampled spatial regions, we used geographic 10-fold cross validation, binning by latitude using data from 2011, 2012, and 2013. Here, the data are split into 10 latitude bins, and each fold is withheld, training a model using all of the data except for the data in the withheld fold. Then, we evaluated the ability of the model to predict the data in the withheld latitude bin, repeating this process 10 times, once for each latitude bin. The results of this 10-fold cross validation provide insight into whether the model can make useful predictions in unsampled spatial locations. Second, to evaluate predictive performance in unsampled years, we withheld the final year of data (2015) as a test set, and trained the model using all of the data from 2011, 2012, and 2013. We note that 2015 provides a particularly challenging out-of-sample test set, because of the presence of a marine heat wave in the Pacific [32]. The combination of these two strategies provide insight into the generalization ability of our model in space and time respectively. We evaluate performance in terms of the coefficient of determination $R^2$:

$$R^2(y, \hat{y}) = 1 - \frac{\sum_{i=1}^{n} (y_i - \hat{y}_i)^2}{\sum_{i=1}^{n} (y_i - \bar{y})^2},$$

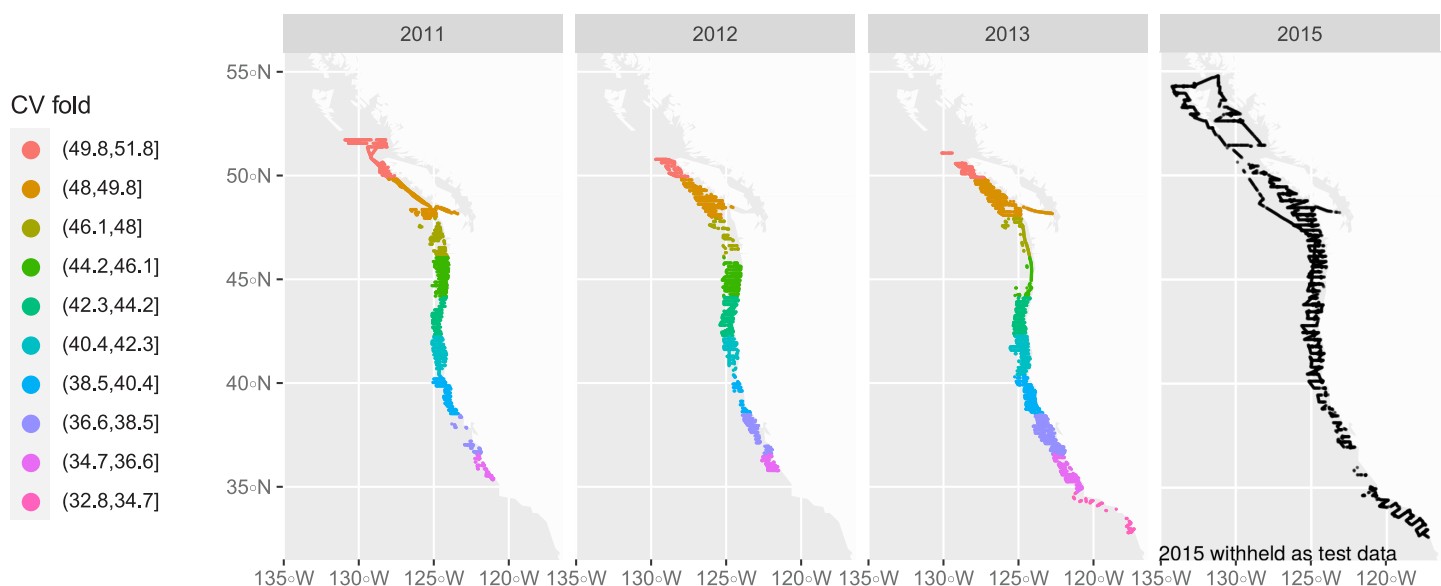

**Fig 1. Acoustic data coverage by year.** Color represents the ten cross validation (CV) folds that correspond to latitudinal bins. Each point represents a spatial location along a trackline where data were collected. The 2015 points are black, and represent a withheld test set that was not used in spatial cross validation.

where $y_i$ is the $i^{th}$ observed value of log NASC, $\hat{y}_i$ is a predicted value, $\bar{y}$ is the sample mean of observed values, and $n$ is the number of observations. This measure of performance can be negative. For ease of visualization we round all negative values to 0 in subsequent discussion, where a 0 value indicates the worst possible value. To understand how predictive performance varied as a function of sonar frequency and depth bin, we compute $R^2$ values for model-frequency-depth bin combinations.

To test temporal forecasting in areas that have been sampled previously, we evaluated the ability of the model to predict the data from 2015 using $R^2$ and with graphical predictive checks. The location of sonar data, latitude bins (for the cross-validation), and test data are shown in Fig 1. All models were fit in R version 3.6.2 using the ranger package [33, 34].

## Results

The performance of the full and reduced models indicate that satellite data may help to predict sonar observations under some circumstances, though the overall performance of all models was relatively low. Across all depth bins and frequencies, holdout $R^2$ values from spatial cross-validation range from 0 to 0.315 (mean = 0.059) for the full model, and 0 to 0.245 (mean = 0.042 for the reduced model). For many of the latitude, sonar frequency, depth bin combinations, performance of both models was quite poor (with many $R^2$ values less than or equal to zero (Fig 2). For the shallow water depth bin, where we expected the satellite data to provide the most useful information, performance of the full model tended to be higher than performance from the reduced model for a subset of the latitude bins, and $R^2$ for the shallow water bin exceeded 0.1 just once south of 40.4 degrees latitude. Notable decreases in predictive performance are apparent in the extreme southernmost and northernmost latitude bins (Fig 2).

Results from the withheld test set containing data from 2015 suggest that prediction in years with no data are generally worse than predictions to new spatial regions in years where data are available. All raw $R^2$ values were less than zero for the full and reduced models (and were set to zero after rescaling as described in the methods). To investigate this, we computed

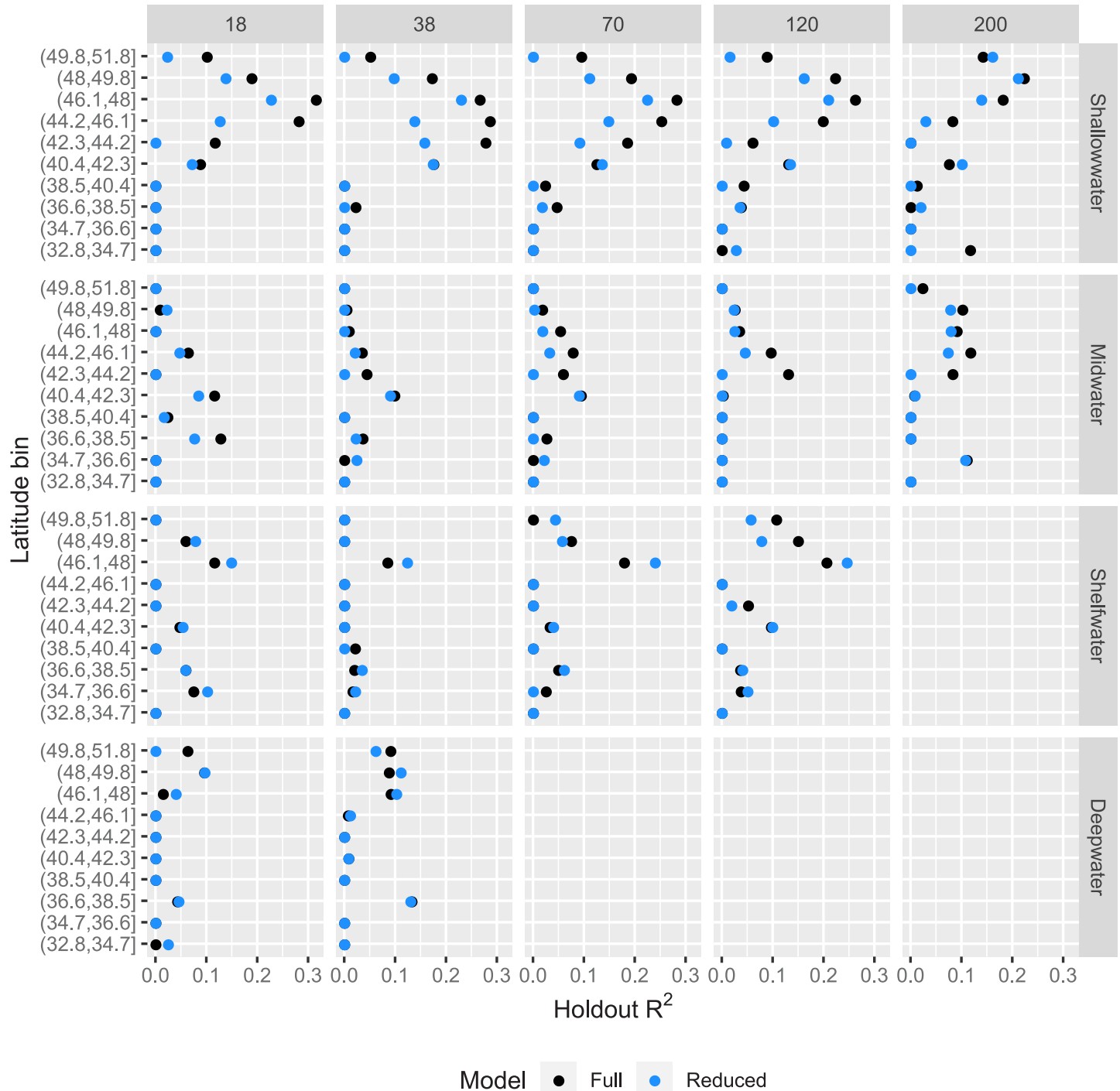

**Fig 2. Spatial cross-validation.** Spatial cross-validation performance comparison for the full model (black) inclusive of satellite observations, and the reduced model (blue) that excludes satellite observations, faceted by depth bin and sonar frequency (kHz). The x-axis displays the $R^2$ value on the withheld latitude bin, and the y-axis displays the latitude bin that was withheld during cross-validation. No results are shown for depth bin/frequency combinations for which the sonar data are known to be unreliable.

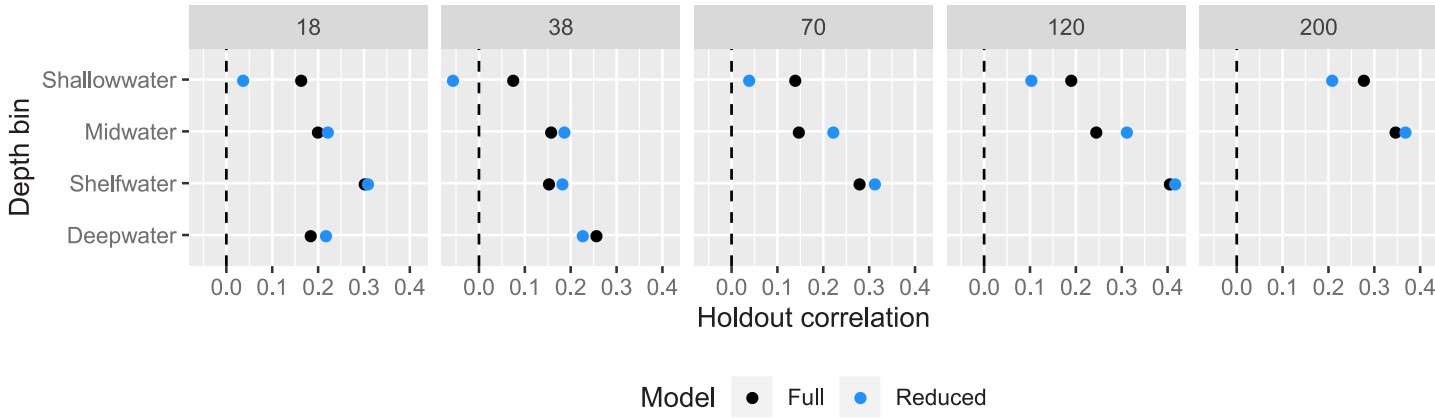

**Fig 3. Withheld 2015 test set correlations.** Pearson correlation coefficients between the true and predicted log NASC values in the 2015 test set, with black representing the full model and blue representing the reduced model. The x-axis represents the correlation values, and the y-axis represents depth bins. The panels represent the sonar frequency (kHz).

correlations between the observed and predicted values by depth bin and sonar frequency, and visualized the relationship between observed and predicted values. In one case, the model predictions were actually negatively correlated with the true values, but all other correlations between the true and predicted values were positive (Fig 3). Notably, in terms of correlation, the full model consistently outperformed the reduced model for the shallow depth bin (Fig 3). However, $R^2$ values were negative or zero despite these positive correlations because of systematic downward bias in model predictions. Specifically, the true average log NASC values were always higher than the predicted averages (Fig 4). Thus, although the model could make predictions about where log NASC would be relatively low or high (hence the positive correlations), bias in these predictions resulted in extremely low $R^2$ values.

## Discussion

We found equivocal evidence for satellite data providing useful information for predicting water-column sonar observations. Based on this exploratory study, it seems that satellite observations are most useful when interpolating spatially, and less useful when extrapolating in space and time. This finding is supported by 1) the difference in out-of-sample predictive performance between the spatial cross-validation and two-years-ahead predictions for the 2015 test set, and 2) the decrease in spatial cross-validation performance in the extreme northern and southern latitude bins, which involve spatial extrapolation.

Overall, sonar observations were hard to predict with high accuracy. The shallowest depth bins had the highest predictive performance, which suggests that information provided by the MODIS data products is more relevant for surface observations than observations in mid to deep water. Spatio-temporal dynamics of fish and plankton are complex and non-stationary, so that for example knowing chlorophyll concentration in a particular location at some time does not necessary provide much information on fish and plankton composition or density at depth. Inclusion of sustained features, such as chlorophyll and sea surface temperature fronts, would be beneficial in future work as these conditions are known to influence the biogeography [35, 36].

Poor performance in the withheld 2015 test set might be a consequence of anomalous conditions in the CCE that resulted from a strong El Niño event that occurred from 2015 into 2016, with a particularly strong Kelvin wave and atmospheric response [37, 38]. Impacts on

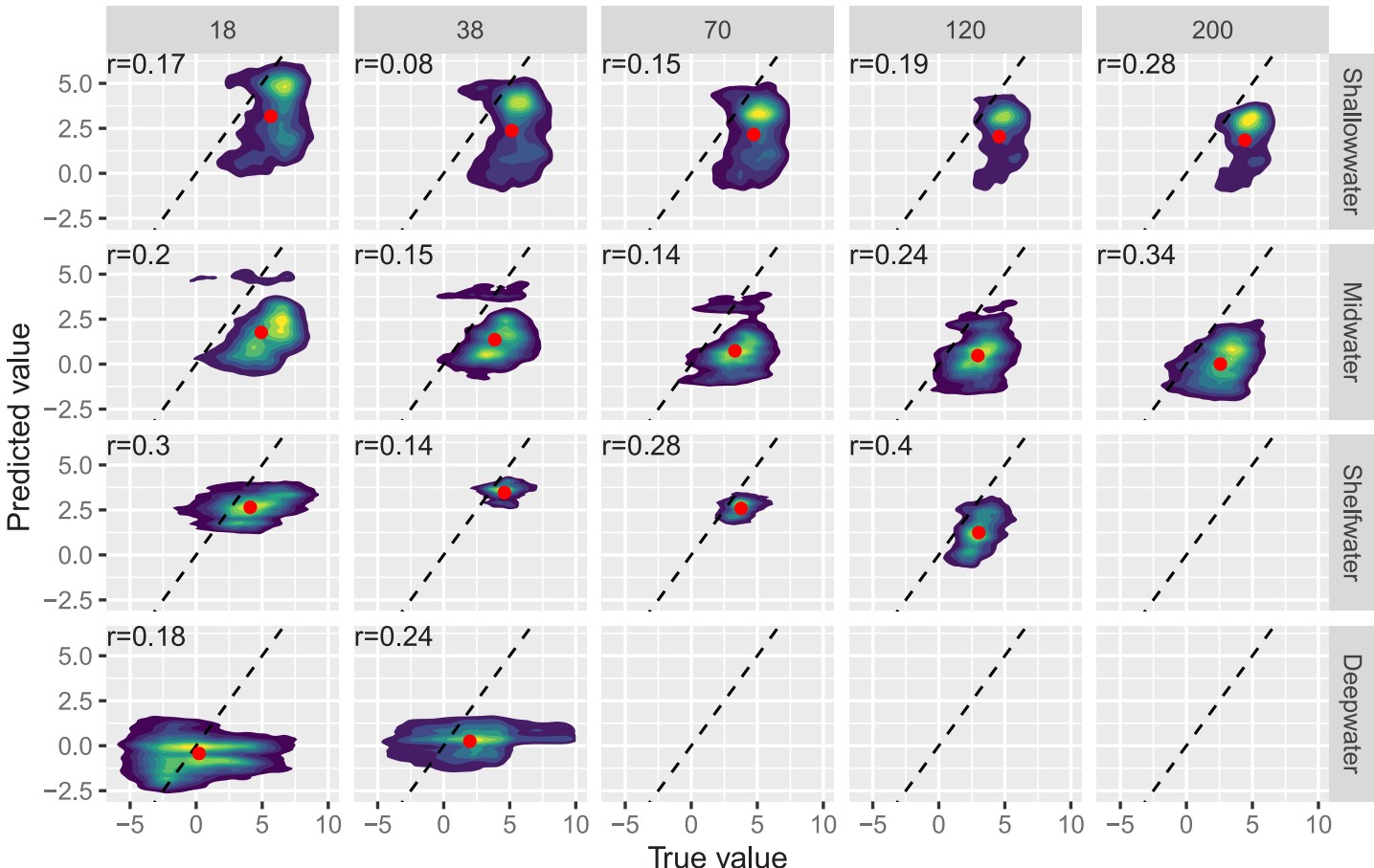

**Fig 4. Observed vs. true values in the test set.** Density surface plots for the relationship between predicted and true (observed) log NASC values in 584,917 observations from the withheld 2015 test set and the full model, by depth bin (panel rows) and sonar frequency in kHz (panel columns). Yellow represents high point density, and dark blue represents low point density. A dashed one-to-one line is shown in every panel, the mean values of true and predicted log NASC are shown as red dots. Red dots far from the dashed line indicate bias in the mean predictions. Pearson correlation coefficients are printed in the upper left of each panel.

the local circulation and ecosystem within the CCE have been explained [39] and reported by *in situ* measurements [40]. In particular this event brought anomalies in both SST and Chlorophyll a over much of the study region [39], such that the data withheld in the 2015 test set contained SST and Chlorophyll a values near or outside the range of values in the 2011-2013 training data. As a result, the predictions for 2015 likely represent a combination of extrapolation forward in time and extrapolation in the space of the model inputs (SST, Chlorophyll a) for any particular spatial location—a difficult task that provides one explanation for the bias observed in the 2015 predictions.

Previous work has predicted *in situ* ocean observations as a function of satellite observations using machine learning. For example, MODIS-derived sea surface temperature and chlorophyll a have been combined with random forests to predict sea surface salinity [41]. Similarly, MODIS satellite observations have also been used to identify harmful algal blooms [42]. Zwolinski et al. [8] developed predictive habitat maps for spawning Pacific sardine following optimal ranges of satellite-derived sea surface temperature, chlorophyll a concentrations, and sea surface height gradients using a generalize additive model. This and follow-on work combined acoustic surveys and *in situ* sampling to verify sardine presence [8, 43]. Generalized additive models were used to model sonar data at 70 kHz as a function of

satellite-derived chlorophyll a, sea surface temperature, spatial location, depth, and distance from shore off the Hawaii Islands [44]. Copeland [44] found latitude, longitude, chlorophyll-a, depth, distance from shore, and sea surface temperature significantly explained roughly 35% of NASC variance. However, a one-to-one comparison with this project is not possible because out of sample predictive performance was not evaluated, and the satellite data were not withheld in a reduced model.

Looking forward, it seems that embedding more mechanistic or process-based structure in an approach like the one developed here might be beneficial. A hybrid approach that combines a parametric dynamical model with a more flexible machine learning model (like a neural hierarchical model) might provide one such approach [45]. A neural hierarchical model would for example provide a framework for specifying a model for a process (e.g., hake migration) and a linked model of data collection (e.g., the location of research vessels). In contrast, an unconstrained machine learning model like a random forest does not account for spatio-temporal mismatch in the trajectory of a vessel collecting sonar data, and the spatio-temporal dynamics of the biological processes of interest.

Additionally, there was potential for geographic misalignment in the spatial aggregation of the sonar data to the spatial resolution of the MODIS product. The sonar data are recorded and stored as irregular point data, and when aggregated to 4 km horizontal bins along the vessel's path, there is no guarantee that this aggregation along the path matches the MODIS grid. In other words, while both data products are aggregated to 4m, the spatial boundaries between 4km sonar bins and 4km MODIS grid cells were not perfectly aligned. Properly aligning the sonar aggregations to the MODIS grid, or that of any other satellite data used for ocean color analysis, could enhance predictive power by ensuring geographic alignment.

One potential limitation in this work is the use of raw explanatory variable values rather than embed information about how the raw values deviate from long term averages. Such anomalies may be more useful but would involve a translation (typically subtracting the mean) of all relevant explanatory variables. That said, anomalies can be seen as allowing the effect of explanatory variables to vary spatially (depending on how they deviate from the mean), and random forests allow the effects of explanatory variables to vary as functions of other covariates such as latitude and longitude. However, the data examined here spans only 5 years and a longer time series of input data and climatology from which to determine anomalous conditions could be informative.

## Conclusion

Taken together, these results indicate that MODIS satellite observations data may be useful for spatial interpolation of marine sonar data in some cases, but may be unreliable for temporal forecasting or spatial extrapolation. We suspect that the lack of performance is a consequence of a mismatch between the observable surface dynamics of sea surface temperature and chlorophyll a with the complex spatio-temporal subsurface dynamics that relate to marine food webs and population dynamics. Three-dimensional circulation models and sustained surface features (fronts) in addition to a longer time series are recommended in future analyses.

## Acknowledgments

The authors thank two anonymous reviewers for their constructive comments that improved this manuscript.

## Author Contributions

**Conceptualization:** Kellie R. Gadeken, Maxwell B. Joseph, Joseph McGlinchy, Kristopher B. Karnauskas, Carrie C. Wall.

**Data curation:** Kellie R. Gadeken, Maxwell B. Joseph, Joseph McGlinchy, Carrie C. Wall.

**Formal analysis:** Kellie R. Gadeken, Maxwell B. Joseph, Joseph McGlinchy.

**Funding acquisition:** Kristopher B. Karnauskas, Carrie C. Wall.

**Investigation:** Kellie R. Gadeken, Maxwell B. Joseph, Kristopher B. Karnauskas, Carrie C. Wall.

**Methodology:** Kellie R. Gadeken, Maxwell B. Joseph, Joseph McGlinchy, Kristopher B. Karnauskas, Carrie C. Wall.

**Project administration:** Maxwell B. Joseph, Joseph McGlinchy, Kristopher B. Karnauskas, Carrie C. Wall.

**Resources:** Maxwell B. Joseph, Carrie C. Wall.

**Software:** Kellie R. Gadeken, Maxwell B. Joseph.

**Supervision:** Maxwell B. Joseph, Joseph McGlinchy, Kristopher B. Karnauskas, Carrie C. Wall.

**Visualization:** Kellie R. Gadeken, Maxwell B. Joseph, Joseph McGlinchy, Carrie C. Wall.

**Writing – original draft:** Maxwell B. Joseph, Joseph McGlinchy, Kristopher B. Karnauskas, Carrie C. Wall.

**Writing – review & editing:** Kellie R. Gadeken, Maxwell B. Joseph, Joseph McGlinchy, Kristopher B. Karnauskas, Carrie C. Wall.

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
