## [Decision Letter · Decision Letter 0]

11 Sep 2020

PONE-D-20-19973

Predicting subsurface sonar observations with satellite-derived ocean surface data in the California Current Ecosystem

PLOS ONE

Dear Dr. Joseph,

Thank you for submitting your manuscript to PLOS ONE. After careful consideration, we feel that it has merit but does not fully meet PLOS ONE’s publication criteria as it currently stands. Therefore, we invite you to submit a revised version of the manuscript that addresses the points raised during the review process.

We look forward to receiving your revised manuscript.

Kind regards,

Emmanuel S. Boss

Academic Editor

PLOS ONE

Journal Requirements:

2. In your Methods section, please provide additional location information of the study area including geographic coordinates for the data set if available.

Additional Editor Comments (if provided):

Dear authors,

Both reviewer found your paper worthy of publication in Plos One. One had some important comments that I believed, if addressed, will improve your manuscript.

All the best, Emmanuel Boss

Reviewers' comments:

Reviewer's Responses to Questions

**Comments to the Author**

1. Is the manuscript technically sound, and do the data support the conclusions?

Reviewer #1: Yes

Reviewer #2: Yes

2. Has the statistical analysis been performed appropriately and rigorously? 

Reviewer #1: Yes

Reviewer #2: Yes

3. Have the authors made all data underlying the findings in their manuscript fully available?

Reviewer #1: Yes

Reviewer #2: Yes

4. Is the manuscript presented in an intelligible fashion and written in standard English?

Reviewer #1: No

Reviewer #2: Yes

5. Review Comments to the Author

Reviewer #1: This paper uses satellite and other simple environmental data to examine the statistical relationships with measures of acoustic backscatter with the goal of determining if surface satellite data can be used to forecast subsurface biological resources. In general, the statistical models with satellite data did no better (and sometimes worse) than those that did not include satellite data. I think it’s worthwhile pointing out that all of the models performed quite poorly, suggesting that the right parameters were not included in the model. To that end, I have made some specific suggestions below. Developing a strong model was not the stated goal of the paper but stronger explanatory power would have provided reassurance that the general approach was valid and a stronger test of the importance of satellite observations.

I also have concerns about the writing style, particularly in the Introduction. There are no topic sentences within most of the paragraphs and transitions are weak or absent. As a result, it is really hard to identify the gaps in our knowledge and the goals of the study on the first read. I’ve made some specific suggestions for modifying the text to address this but these should not be considered exclusive.

Line 13: I think it’s worth mentioning early that most ship-based echosounder systems miss the upper 5-10 m of the water column because of the depth of the hull, transducer ringdown, and other factors.

Line 17: The transition here is really abrupt. There is no topic sentence that connects the ideas about remote sensing to the measurements presented. What is the take away point here? It needs to be stated up front

Line 28: Similar to above, there is not a clear connection here to previous ideas. Have remote sensing tools been used in the California Current? Why are you focusing on the CCE for this study?

Line 39: Again, there is no topic sentence. Also, I’m not sure which fish are found to 300 and which 100 m since three species are referenced in this line.

Line 48: This paragraph, with modifications, should come before the description of the CCE.

Line 133: I think it’s a stretch from the reference to say that myctophids in the CCE reflect strongly at low frequencies. At the very least, the statement should be qualified.

146: How much does the optical column contributing to MODIS vary?

Line 136: Why was only sea surface color utilized? Why not include other satellite derived data such as sea surface height and temperature? Edited to add that I see at least SST was used as well in Table 1 but it was not included in the description of the methods.

Line 153: Instead of just including ocean depth, bottom slope or some other measurement of structure would probably be more informative.

Conclusions: I was surprised not to see any discussion of lags between chlorophyll production and fish production which could lead to the mismatches observed. It would be interesting to test this using satellite data derived from a period before the acoustic observations to see if there was a better fit. I think it also bears mentioning that previous attempts to use satellite data to predict animal distributions have largely focused on individual species (like sardine) where physiological limitations and foraging condition preferences are known. Looking at the grab bag that NASC is mixes across those mechanisms and species, clouding the picture. In addition, Chlorophyll is a measure of standing stock, not production which further complicates the interpretation.

Reviewer #2: The manuscript Predicting subsurface sonar observations with satellite-derived ocean surface data in the California Current Ecosystem describes an attempt to predict acoustic backscatter from a diverse community of marine organisms using remotely-sensed information. From the get go, I would not expect much success on that endeavor, considering that acoustic backscatter is affected by a myriad of factors, including community structure and their sizes and anatomies, all of which are changing seasonally and annually. However, the authors’ approach and analysis is sound, and one that I particularly appreciate. The cross-validation approaches utilized are adequate to test the predictive performance of the models and should be welcome in the literature. Moreover, unlike many others, the authors did not try to oversell their findings or presented conclusions not supported by the data; the speculations and suggestions into future research avenues are sensible. Overall, despite not yielding what I would consider a “positive” result, this paper stands on its good quality and would certainly serve the scientific community as a blueprint for other studies applying the same type of analyses. In conclusion, I recommend the paper to be published with minor modifications.

Minor comments:

Line 31-33: Aggregations of euphausiid species in the CCE, namely North Pacific krill (Euphausia pacifica), are typically found less than 30 m depth during the night and between 50-100 m depth.

Thysanoessa spinifera is very abundant too in the CCE, and there are many other minor species that could be important at a local level.

Line 82: On-axis [7] beam-pattern measurements were also taken.

Please remove or rephrase. Beam pattern refers to the sensitivity of transmission and reception as a function of angle beyond the axis.

6. PLOS authors have the option to publish the peer review history of their article (what does this mean?). If published, this will include your full peer review and any attached files.

Reviewer #1: No

Reviewer #2: No

---

## [Author Response · Author response to Decision Letter 0]

31 Dec 2020

Thanks for the constructive reviews - our point-by-point responses are in the attached response to reviewer document.

---

## [Editor Report · Decision Letter 1]

5 Jan 2021

PONE-D-20-19973R1

Predicting subsurface sonar observations with satellite-derived ocean surface data in the California Current Ecosystem

PLOS ONE

Dear Dr. Joseph,

Thank you for submitting your manuscript to PLOS ONE. After careful consideration, we feel that it has merit but does not fully meet PLOS ONE’s publication criteria as it currently stands. Therefore, we invite you to submit a revised version of the manuscript that addresses the points raised during the review process.

We look forward to receiving your revised manuscript.

Kind regards,

Emmanuel S. Boss

Academic Editor

PLOS ONE

Additional Editor Comments (if provided):

Dear authors,

I am satisfied with the changes you have made in response to the reviewers' comment.

I have, however, several comments that I feel, if addressed, will make your work more significant.

1. You state: 'Additionally, there was potential for geographic inaccuracies in the aggregation of

the sonar data to the spatial resolution of the MODIS product. The sonar data are

recorded and stored as irregular point data, and when aggregated to 4 km to match the

MODIS products there was no guarantee that the aggregation window matched exactly

that of a single MODIS grid cell.'

Can you please elaborate?

I believe the georeferencing of MODIS is quite good as well as those from your GPS.

Do you imply a problem with the fact that the time measurements were taken at is not the same as the satellite pass which may result in advection of features? If yes, please say so clearly.

The way your explanation is written suggests that maybe it is an issue of subgrid variability (satellite product you use average a 4x4km piece of ocean). Do you have evidence of such patchiness?

BTW, you can obtain chlorophyll at Level 2 at 1x1km grid. No reason to limit yourself to Level 3. You can also use inline sensors, if available, to assess sub-grid variability in some properties.

2. You do not mention ship noise as a possibility for surface acoustics not matching. Did you use a 'quiet' vessel? Is there reason to believe that there may be a response to the vessel's noise?

I am not an acoustician and since the reviewers have not raised this issue, it is only a comment.

3. Please do not call the z1% product 'euphotic depth'. It is an anachronism (I know NASA still uses that, but not for much longer).

Primary productivity is limited by the absolute intensity of light, not relative to a surface value that changes throughout the year. Keep it as the 1% light level. People will know exactly what you mean.

All the best, Emmanuel
---

## [Author Response · Author response to Decision Letter 1]

18 Feb 2021

Please see the response to reviews in the uploaded file.

---

## [Editor Report · Decision Letter 2]

24 Feb 2021

Predicting subsurface sonar observations with satellite-derived ocean surface data in the California Current Ecosystem

PONE-D-20-19973R2

Dear Dr. Joseph,

We’re pleased to inform you that your manuscript has been judged scientifically suitable for publication and will be formally accepted for publication once it meets all outstanding technical requirements.

Kind regards,

Emmanuel S. Boss

Academic Editor

PLOS ONE

Additional Editor Comments (optional):

Dear authors,

I am satisfied with your revisions.

In future work you may want to aggregate the data into the satellite grid cells to avoid the possible issue of mismatch.

The information on how to do it is in the public domain.

Best, Emmanuel
---

## [Editor Report · Acceptance letter]

10 Aug 2021

PONE-D-20-19973R2 

Predicting subsurface sonar observations with satellite-derived ocean surface data in the California Current Ecosystem 

Dear Dr. Joseph:

I'm pleased to inform you that your manuscript has been deemed suitable for publication in PLOS ONE. Congratulations! Your manuscript is now with our production department. 

Kind regards, 

on behalf of

Dr. Emmanuel S. Boss 

Academic Editor

PLOS ONE